

# Boron-doping of cubic SiC for intermediate band solar cells: a scanning transmission electron microscopy study

### In memory of Professor Bengt Svensson

Patricia Almeida Carvalho[1,2⋆], Annett Thøgersen[1], Quanbao Ma[3], Daniel N. Wright[4], Spyros Diplas[1,5], Augustinas Galeckas[3], Alexander Azarov[3], Valdas Jokubavicius[6], Jianwu Sun[6], Mikael Syväjärvi[6], Bengt G. Svensson[3] and Ole M. Løvvik[1,3†]

**1** SINTEF Materials Physics, Oslo, Norway
**2** University of Lisbon, Instituto Superior Tecnico, Lisbon, Portugal
**3** University of Oslo, Department of Physics, Oslo, Norway
**4** SINTEF Instrumentation, Oslo, Norway
**5** University of Oslo, Department of Chemistry, Oslo, Norway
**6** Linköping University, Department of Physics, Chemistry and Biology, Linköping, Sweden

⋆ patricia.carvalho@sintef.no † olemartin.lovvik@sintef.no

## Abstract

Boron (B) has the potential for generating an intermediate band in cubic silicon carbide (3C-SiC), turning this material into a highly efficient absorber for single-junction solar cells. The formation of a delocalized band demands high concentration of the foreign element, but the precipitation behavior of B in the 3C polymorph of SiC is not well known. Here, probe-corrected scanning transmission electron microscopy and secondary-ion mass spectrometry are used to investigate precipitation mechanisms in B-implanted 3C-SiC as a function of temperature. Point-defect clustering was detected after annealing at 1273 K while stacking faults, B-rich precipitates and dislocation networks developed in the 1573 - 1773 K range. The precipitates adopted the rhombohedral $B_{13}C_2$ structure and trapped B up to 1773 K. Above this temperature, higher solubility reduced precipitation and free B diffused out of the implantation layer. Dopant concentrations of $10^{19}$ at.cm$^{-3}$ were achieved at 1873 K.

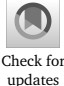
## 1   Introduction

An intermediate band (IB) in the energy band gap of a semiconductor allows photons with lower energy than the band gap to excite electrons from the valence to the conduction band, increasing the photocurrent generated [1]. Due to the potential for enhanced efficiency in energy conversion, intermediate-band solar cells are promising candidates for the next generation of photovoltaic devices. Theoretical efficiencies of 63% have been estimated for IB solar cells under concentrated sunlight [1], a value considerably higher than the maximum efficiency of 40% expected for conventional single p-n junctions [2].

Realization of IB solar cells has faced challenges in finding a semiconductor/dopant system with band gap in the 1.9 - 2.5 eV range and suitably positioned intermediate band. Cubic silicon carbide (3C-SiC) has a nearly ideal band gap of 2.36 eV at room temperature, combined with useful electronic properties [3], and boron (B) has been proposed to form a deep acceptor level 0.7 eV above the valence edge ($E_v$) of SiC [4]. Thus, B-doped 3C-SiC is a promising absorber system for highly efficient photovoltaic devices.

The hydrogen-like model with the values of permittivity and effective hole mass reported for 3C-SiC [5] estimates $10^{19}$ - $10^{20}$ at.cm$^{-3}$ as the minimum concentration of shallow acceptors for their energy levels to merge into impurity bands. However, the two acceptor levels associated with B in SiC are rather deep (with positions at $\sim E_v + 0.3$ eV and $\sim E_v + 0.7$ eV [4], which may push the required concentration to even higher values, emphasizing the need for assessing and controlling the precipitation behavior in the system. Limited work has been carried out on doping 3C-SiC with B and the literature available is essentially theoretical due to the difficult synthesis of high quality crystals [6–9]. Recent improvements in growth techniques are renewing experimental efforts on bulk 3C-SiC [10, 11], and the optical activity deduced so far from absorption and emission spectra of B-implanted 3C-SiC indicates, indeed, IB behavior [12–14]. Nevertheless, ascertaining the electronic configuration of the system demands heavily doped and structurally sound 3C-SiC samples, and thus processing conditions above the B solvus.

In the present work, sublimation-grown 3C-SiC crystals were implanted with B at elevated temperatures and the samples were subsequently annealed in the 1273 - 2073 K temperature range. Crystalline defects and precipitation mechanisms were investigated by scanning transmission electron microscopy (STEM) using variable collection angles to evidence specific local features. Correlation of the microstructural observations with secondary-ion mass spectrometry (SIMS) data was used to evaluate B solubility.

## 2   Experimental

High-quality 3C-SiC single crystals with an area of $\sim 7 \times 2 \, \text{mm}^2$ and thickness $> 200 \, \mu\text{m}$ were grown on 4H-SiC substrates off-oriented 4° from [0001] by sublimation epitaxy, resulting in an exposed surface close to $(111)_{\text{3C-SiC}}$ (details on the process can be found elsewhere [10]). Due to the processing conditions, nitrogen (shallow acceptor) can be present in concentrations

up to $10^{17}\,\mathrm{cm}^{-3}$ turning the as-grown material into an n-type semiconductor [12]. Hall-effect measurements were used to attest the n-type character of the 3C-SiC single crystals and determine their carrier density ($\sim 10^{16}\,\mathrm{cm}^{-3}$) and resistivity ($\sim 18 \pm 1\,\Omega.\mathrm{cm}$) [12].

Implantation with $^{11}\mathrm{B}^{+}$ ions was carried out at elevated temperatures (673 and 773 K) along a direction close to $(111)_{3C\text{-SiC}}$ using multiple energies (100 to 575 keV) with a total dose of 4 - 13 × $10^{16}$ atoms.cm$^{-2}$ to form box-like concentration profiles of about 1, 2 and 3 at.% B (used henceforth to designate the concentration in the material). The profiles extended about 600 nm in depth either directly below the free surface or buried with a start at $\sim 300$ nm. Post-implantation annealing was carried out in the 1273 - 1973 K temperature range for 3600 s and at 2073 K for $1.4 \times 10^{4}$ s. Combinations of different B concentration level and annealing temperature/time have been investigated to infer precipitation and defect formation trends within a large window of processing parameters. Prior to annealing, the samples were protected by a pyrolized resist film (carbon cap) after native oxide etching. The pyrolysis was performed in forming gas at 1173 K for 6 - 9 × $10^{2}$ s and the carbon cap was removed by dry thermal oxidation at 1173 K for 2 - 3 × $10^{2}$ s before the measurements. The crystallinity of the as-grown, as-implanted and annealed samples was attested by grazing incidence X-ray diffraction and Rutherford backscattered spectroscopy [13].

Absence of extended structural defects in the sublimation-grown 3C-SiC single crystals was confirmed by conventional transmission electron microscopy (TEM) prior to the B implantation. The observations were performed close to $\langle 1\bar{1}0 \rangle$ zone axes for easy detection of the typical stacking faults on {111} planes. The implanted and annealed samples were investigated by TEM and by annular bright-field (ABF), low-angle annular dark field (ADF) and high-angle annular dark field (HAADF) STEM. The microscopy work was performed with a DCOR Cs probe-corrected FEI Titan G2 60-300 instrument with 0.08 nm of nominal spatial resolution. Chemical information was obtained by X-ray energy dispersive spectroscopy (EDS) with a Bruker SuperX EDS system, comprising four silicon drift detectors, and by electron energy loss spectroscopy (EELS) with a GIF Quantum 965 EELS Spectrometer. TEM sample preparation was performed by focused ion beam with Ga$^{+}$ ions accelerated at 30 kV using a JEOL JIB 4500 multibeam system. Lattice images were indexed using fast Fourier transforms (FFT) and strain was evaluated by geometric phase analysis (GPA) using the FRWRtools plugin [15] implemented in Digital Micrograph (Gatan Inc). The crystallographic orientations between precipitate variants and the 3C-SiC matrix were investigated using crystallographic data retrieved from the literature [16]. The Carine Crystallography 3.1 package [17] was employed to associate stereographic projections to the respective lattices. Phase diagrams of the B-C-Si system [18–20] have been used to interpret the microstructural configurations.

Boron concentration across sample depth was measured by SIMS using a Cameca IMS 7f microanalyzer with a primary sputtering beam of 10 keV $O_2^{-}$ ions rasterized over $150 \times 150\,\mu\mathrm{m}^2$ with lateral resolution of 1 $\mu$m and detection of $^{11}\mathrm{B}^{+}$ secondary ions. Absolute values of B concentration were obtained via calibration with ion-implanted reference samples. The depth of the sputtered crater was measured using a Dektak 8 stylus profilometer and a constant erosion rate was assumed for conversion of sputter time to depth.

## 3 Results and Discussion

### 3.1 Microstructure evolution

Cross-sectional STEM images obtained directly after implantation and after annealing at 1273 K are shown in figure 1 together with the corresponding concentration profiles obtained by SIMS. At low collection angles, the implanted regions exhibited homogenous but distinctive

contrast compared to the underlying material (figure 1(a)). This is expected to result from displaced Si atoms rather than from the introduction of the weakly-scattering B atoms [21] (or from displacing the also weakly-scattering C atoms). In silicon, scattering induced by lattice displacement around a single substitutional B atom peaks at 40 mrad and several atoms superimposed along the atomic columns can add up to significant scattering [22]. Boron has been proposed to preferentially substitute Si in 4H-SiC [23]. However, boron-nitrogen donor-acceptor transitions (DAT) indicate that the behavior is more complicated, with B presumably also substituting C [24,25], and possibly forming a complex in which a B atom in a Si site is paired with a carbon vacancy ($B_{Si}$-$V_C$), the so called deep boron-related D-center at $\sim E_v + 0.7$ eV [26,27]. Nevertheless, given the differences in scattering cross-section between Si and the other atoms/point defects, the electron scattering behavior of B-implanted 3C-SiC is expected to be similar to that of B-implanted Si. At high collection angles (98 - 200 mrad), as the electrons scattered to lower angles by the point defects are filtered out, no significant contrast between the implanted layers and the underlying material is discernible (other than the one attributable to continuous variation of the sample thickness, figure 1 (b)). After annealing at 1273 K, local contrast variations uniformly distributed across the implanted layer were detected at low collection angles (figure 1 (c)), while no significant changes occurred in the concentration profile (figure 1(d)).

Magnified images of the contrast variations observed in figure 1 (c), as obtained simultaneously with the ABF and HAADF STEM detectors, are presented in figure 2 (a) and (b), respectively. The local variations detected at low collection angles in ABF (or ADF, see figure 1 (c)) are absent in the HAADF images produced with electrons scattered to high angles (figure 2 (a) vs (b)). Due to the greater contribution from coherent scattering at low/moderate collection angles, ABF and ADF images are more sensitive to defocus and thickness variations, as well as to the presence of strain, than HAADF images [22]. However, the fine variations observed in figure 2 (a) are not expected to originate from a tilted or wavy specimen, or from local differences in thickness. Furthermore, any changes in the electron exit wave resulting from the presence of amorphous layers on the top and lower surfaces of the TEM sample would not be confined to the implanted layer (see uniform contrast below the implanted layer in figure 1 (c)). Geometric phase analysis of both ABF and HAADF images was carried out with 111 and 002 passbands to investigate whether such contrast variations resulted from strain (see figure 2 (c) and (d)). Indeed, apparent strain localization compatible with the compressive/tensile strain fields characteristic of edge dislocations was found in the ABF images (see yellow arrows in figure 2 (c)). However, these defect configurations were not confirmed by GPA performed on the HAADF images (figure 2 (d)). Closer inspection of the atomically resolved ABF images revealed sharp contrast reversals along (111) planes (see magnified inset in figure 2 (e)), which do not correspond to consistent displacements of the heavier Si columns, as evidenced in figure 2 (f). Therefore, the contrast variations at low collection angles result probably from amplitude and phase changes induced on the electron wave by clustering of the point defects generated by implantation.

After annealing at 1673 K, a large density of stacking faults was observed on (111) planes parallel to the surface, with lower fault density present on the concurrent {111} planes (see straight lines in figure 3 (a) and (b)). The absence of these defects in the samples annealed at lower temperatures (see Figures 1 (c) and 2 (a)) implies that additional thermally activated lattice rearrangements occurred at 1673 K. A mottled contrast compatible with the presence of clusters or precipitates was barely discernible in BF TEM (see arrows in figure 3 (b)), while no significant changes were detected in the B concentration profiles (figure 3 (c)).

Different types of contrast were observed after heat treatment at even higher temperatures, as shown in figure 4. Extended structural defects, such as dislocations and stacking faults, separating mosaics with slightly different crystallographic orientations, were present



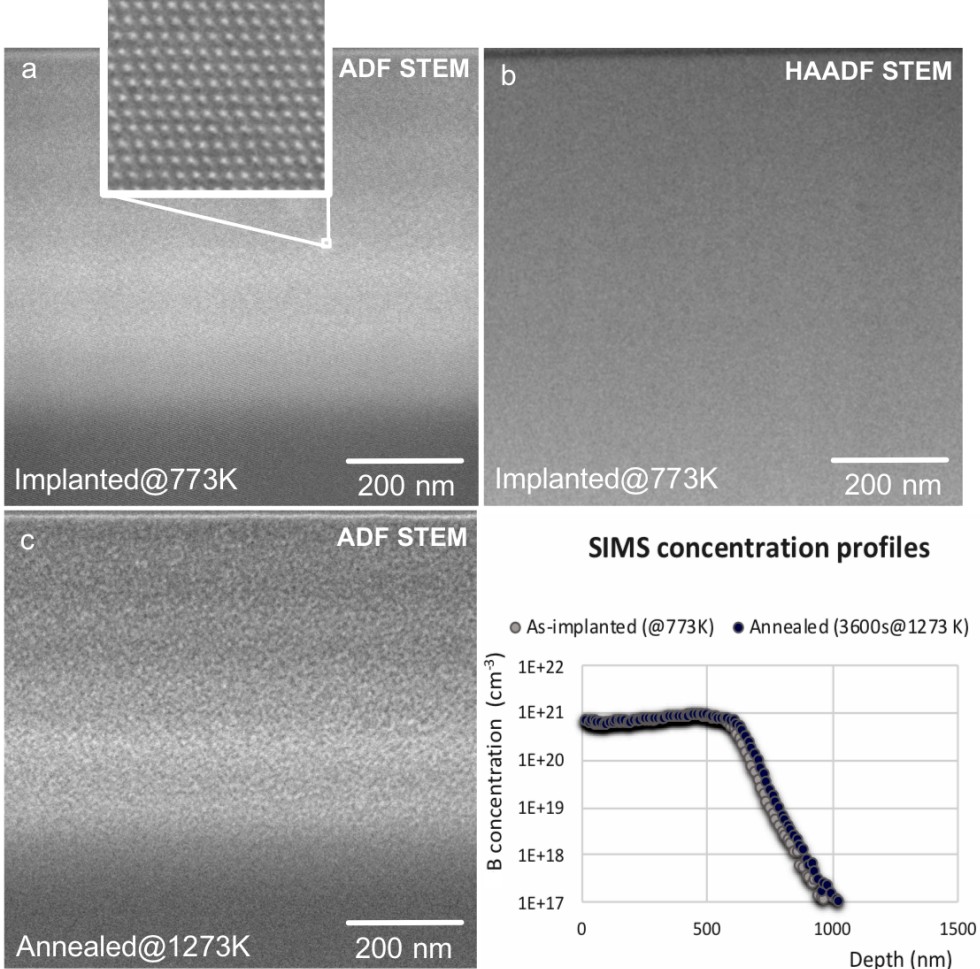

Figure 1: Annular dark-field STEM images of a sample implanted with 1 at.% B at 773 K. Same region observed with (a) 48 - 200 and (b) 99 - 200 mrad collection angles. The sample surface corresponds to the images top. (c) Annular dark-field STEM image of the same layer after annealing at 1273 K for 3600 s obtained with 22 - 98 mrad collection angles. Convergence angle: 21 mrad. (c) SIMS concentration profiles measured from the samples in (a,b) and (c).

in ABF and ADF images after annealing at 1773 K (figure 4 (a,d) and (b,e), respectively), whereas low-mass precipitates were the dominating feature in HAADF images (figure 4 (c,f)). Annealing at 1873 K resulted in lower defect density but large precipitates pinned curved dislocations (figure 4 (g) to (i)). The lower precipitate density and larger precipitate size at 1873 K (2 at. % B) compared to 1773 K (3 at. % B) points to lower nucleation rate, as expected for lower undercooling and/or lower supersaturation (lower initial concentration and higher solubility). Ostwald ripening after precipitation may also have contributed to the increased particle size [28].

The microstructural changes occurring during the heat treatments are summarized in figure 5 (a) using ADF images (where both extended structural defects and low-mass precipitates present distinctive contrast) for different concentration and annealing conditions. The observations suggest that precipitation was controlled by diffusion at the lower annealing temperatures and by driving force (supersaturation and undercooling) at the higher temperatures, with highest rate at around 1773 K (see also Figure 4 (f)). Quantitative evaluation of precipitation parameters was complicated by the diffusive fluxes out of the implanted layer, which promoted

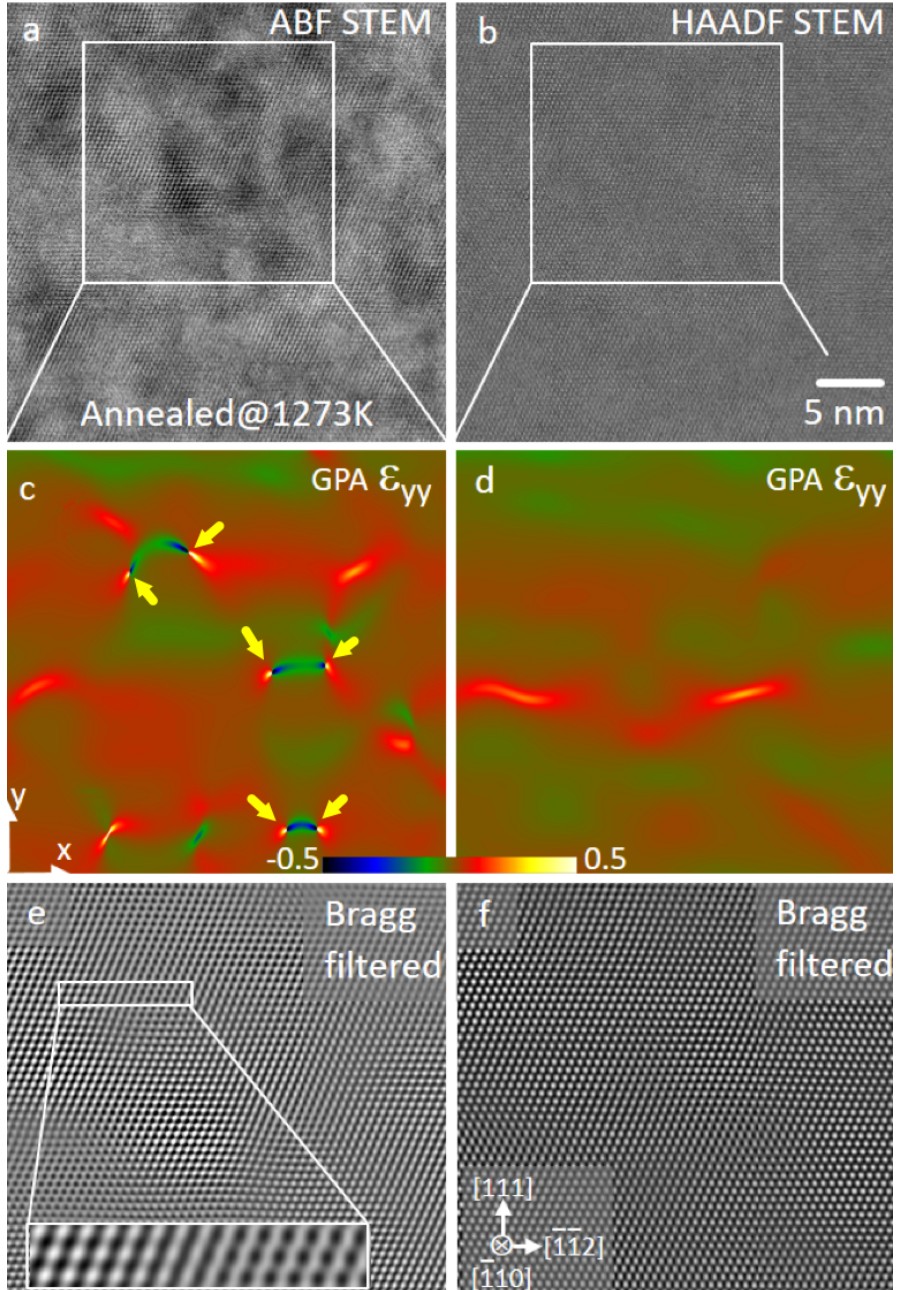

Figure 2: (a) and (b) ABF and HAADF STEM images of 3C-SiC implanted with 1 at.% B at 773 K and annealed at 1273 K. The images were obtained simultaneously from the same region with a convergence angle of 31 mrad and collection angles of respectively, 11 - 21 mrad and 99 - 200 mrad. (c) and (d) Strain maps obtained with the 111 and 002 reflections from the insets in the (a) and (b) raw images. The x and y directions are indicated in (c) and yellow arrows point to apparent compressive/tensile strain fields characteristic of edge dislocations. (e) and (f) Bragg-filtered images of (a) and (b) insets.

precipitate dissolution. Nevertheless, a qualitative description is schematized in (figure 5 (b)).

Complete elimination of defects was not accomplished in the conditions studied, since Lomer-Cottrell locks [29] and precipitates stabilized by extended defects were detected after annealing at 2073 K for $1.4 \times 10^4$ s (see arrows in figure 5 (a)). Structural defects, such as

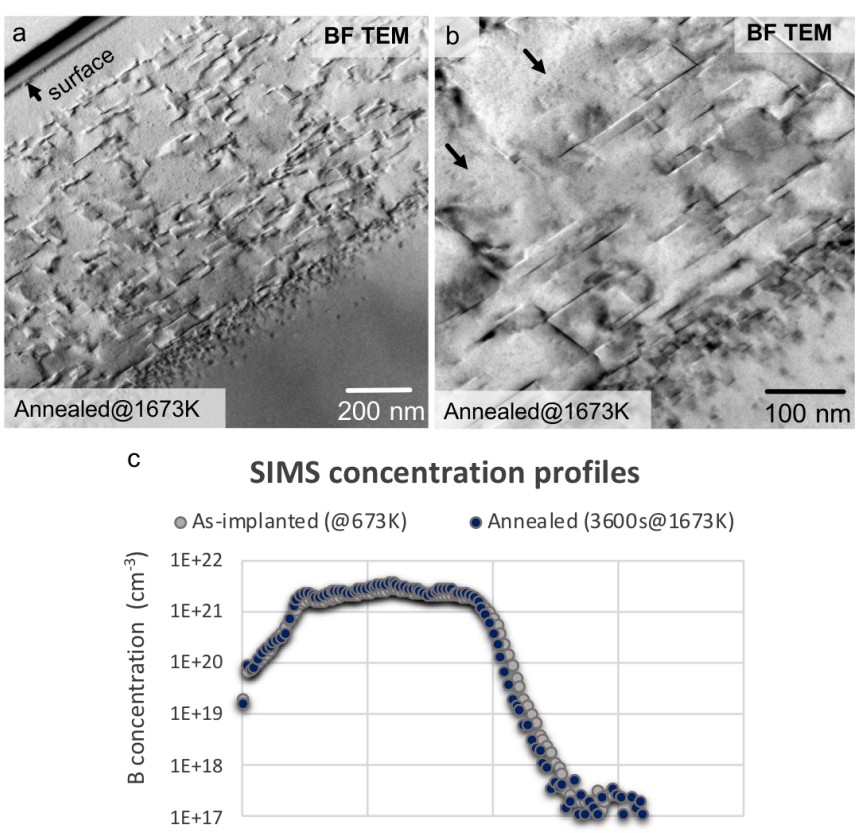

Figure 3: (a) and (b) Bright-field TEM images at different magnifications of a 3 at.% B layer implanted at 673 K and annealed at 1673 K for 3600 s. The arrows in (b) indicate clusters/precipitates and the straight lines are stacking faults parallel to {111} planes. (c) SIMS concentration profiles measured in the as-implanted and annealed states.

precipitates, stacking faults and dislocations, are inherently associated with electronic transitions and can contribute to the optical and electrical activity of the material, potentially masking/mimicking IB behavior in absorption/emission spectra.

## 3.2 Precipitate phase

Despite the low fluorescence yield of B (EDS) and the overlapping of the B-K edge with the strong Si-L2,3 edge (EELS), the local spectroscopy techniques demonstrated that the low mass precipitates were rich in B (red tint in figure 6). The boride precipitates tended to adopt platelet morphologies but their atomic structure was not easily resolved due to the intrinsically weaker B scattering and the strong contribution of the embedding 3C-SiC matrix (see figure 7 (a) to (c)).

Stacking faults running in the matrix often terminated at precipitates (see arrows in figures 4 and 7), in some instances in association with Lomer-Cottrell locks (figure 7 (d)). Since stacking faults were absent both in as-implanted samples and after annealing at 1273 K (figures 1 and 2), these defects were generated during precipitate growth, probably through a stress relaxation mechanism. Therefore, precipitation has additional deleterious implications on the overall structural quality of the B-doped 3C-SiC crystals.

Large precipitates presented facets parallel to $\{111\}_{3C-SiC}$ and $\{002\}_{3C\text{-}SiC}$ when imaged

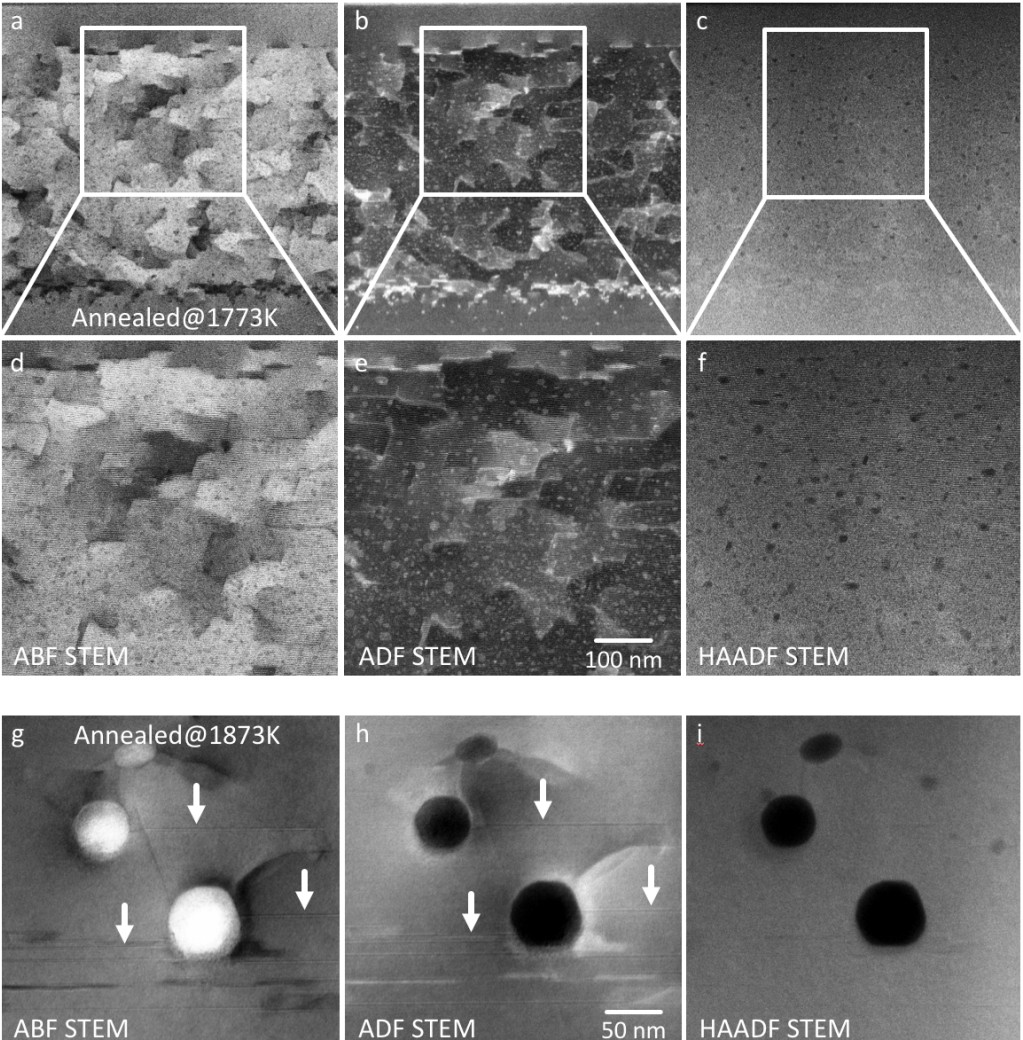

Figure 4: (a), (b) and (c) ABF, ADF and HAADF STEM images obtained simultaneously with a convergence angle of 22 mrad from a layer implanted with 3 at.% B at 673 K and annealed at 1773 K. (d) to (f) Magnified details of (a) to (c). (g), (h) and (i) ABF, ADF and HAADF STEM images obtained simultaneously with a convergence angle of 31 mrad from a sample implanted with 2 at.% B at 673 K and annealed at 1873 K. The arrows indicate stacking faults in the SiC matrix ending at a precipitate. Collection angles for ABF: 11 - 21 mrad, ADF: 22 - 98 and HAADF: 99 - 200 mrad.

along $\langle 1\bar{1}0\rangle_{\text{3C-SiC}}$ and exhibited high density of planar defects, as illustrated in figure 8 for two overlapping platelets with different crystallographic orientation. The symmetry and lattice spacing are consistent with the rhombohedral $B_{13}C_2$ cell [16]. Variants of the $(0001)_{B_{13}C_2} // \{111\}_{\text{3C-SiC}}$ and $\langle 1\bar{1}00\rangle_{B_{13}C_2} // \langle 1\bar{1}0\rangle_{\text{3C-SiC}}$ orientation relation have been found, as depicted in figure 9. Analysis of the additional spots and streaks present in the Fourier transforms of the high-resolution images revealed that the planar defects lied on first-order pyramidal planes, i.e., $\{1\bar{1}01\}_{B_{13}C_2}$ [30–32].

Boron carbide is a well-known semiconductor with electronic properties dominated by hopping-type carrier transport [33]. This is relevant in the context of the present investigation because the optical behavior of B-doped 3C-SiC may be masked by the presence of $B_{13}C_2$ precipitates acting as embedded quantum dots with electronic transitions in spectral ranges similar to those expected for an IB in 3C-SiC. Luminescence peaks exhibited by boron carbide

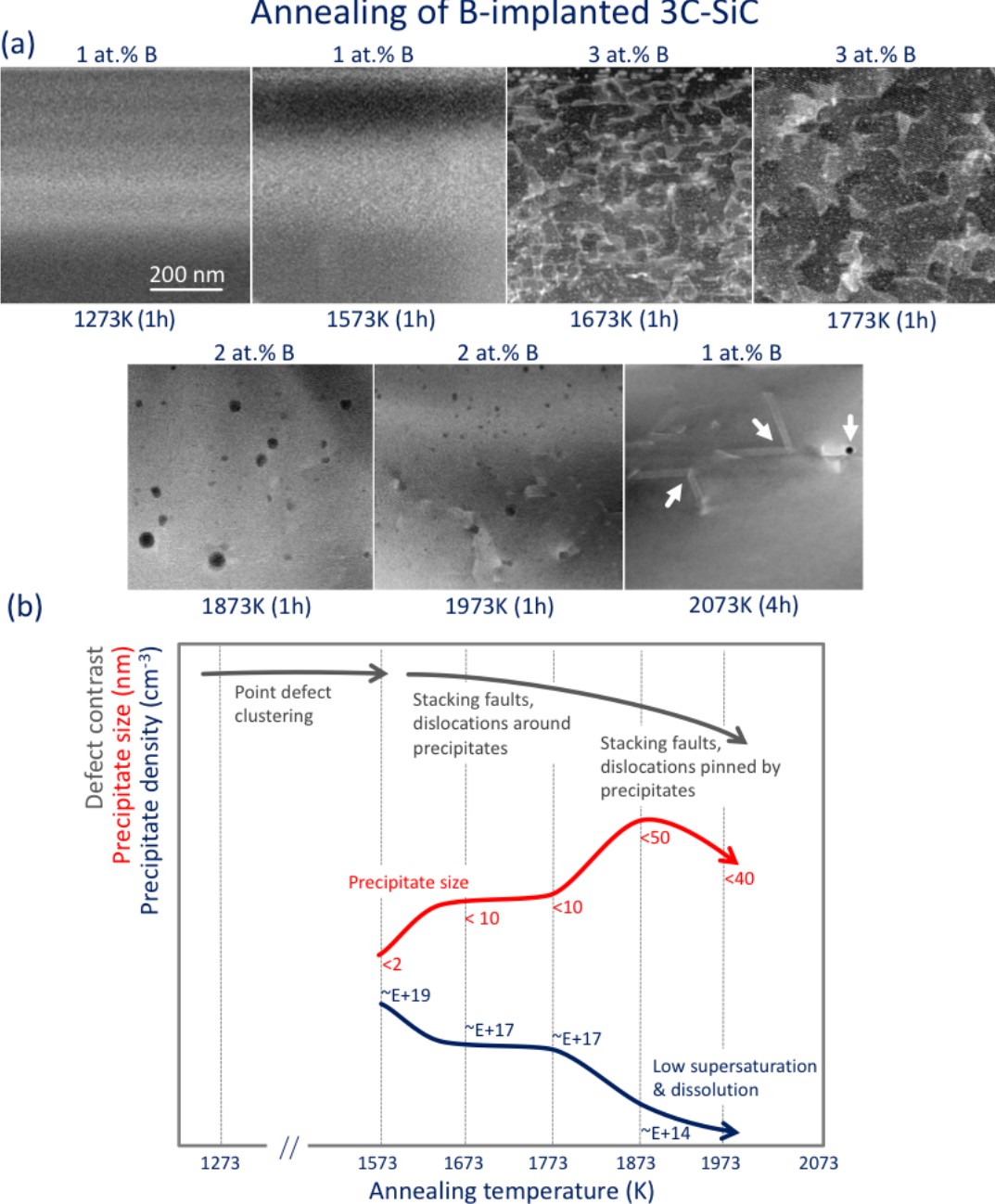

Figure 5: (a) ADF images of B-implanted 3C-SiC samples (same magnification) for different B concentration level and annealing temperature/time, as indicated for each image. The arrows in the image of the sample annealed for 4 h at 2073 K point to Lomer-Cottrell locks (left and center) and to a locked defect configuration involving a large precipitate and a planar defect (right). (b) Qualitative illustration of the microstructural evolution in the implanted layer with annealing temperature. The precipitate density was estimated from the initial concentration and approximate precipitate size assuming spherical shape.

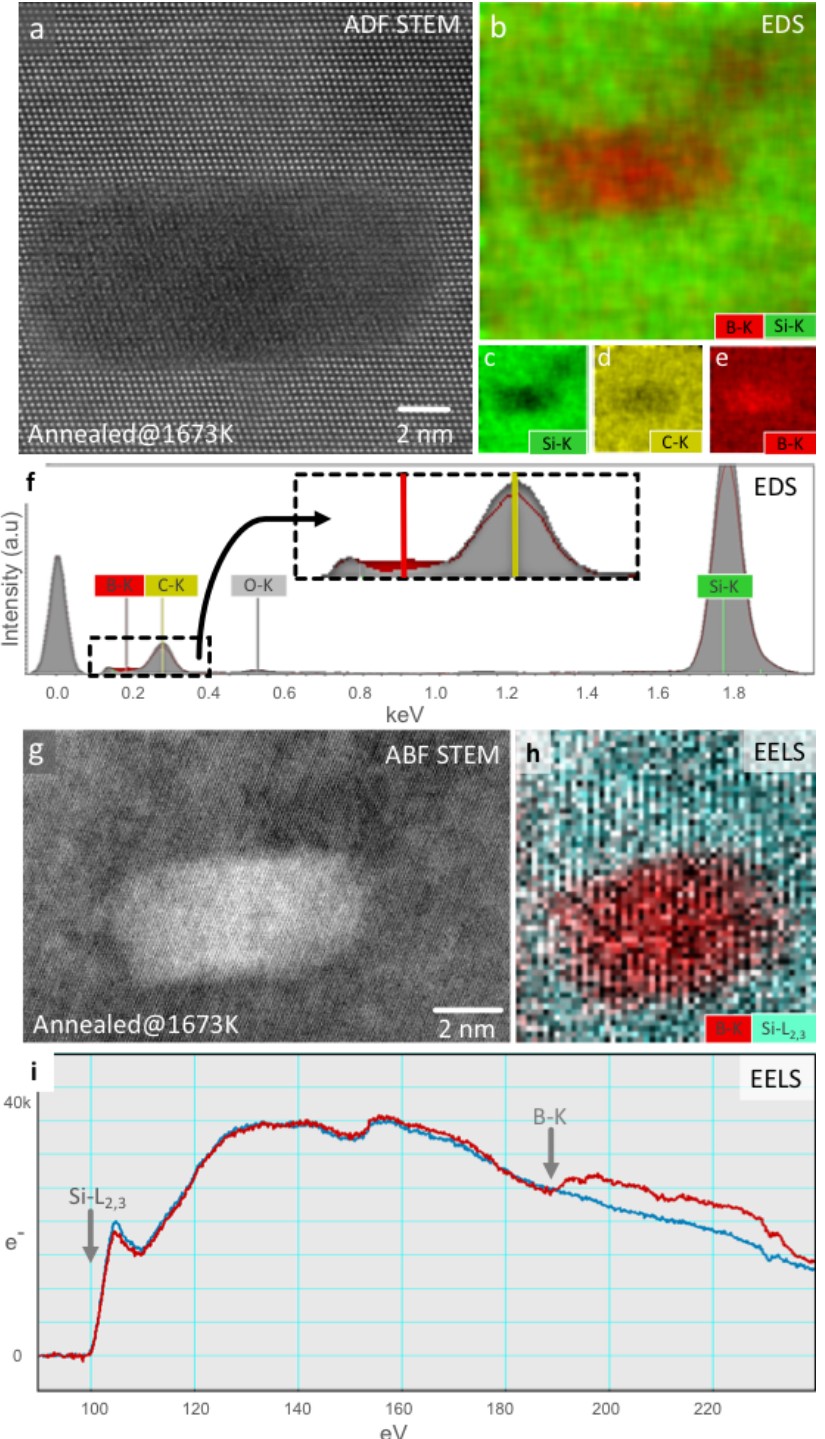

Figure 6: Sample implanted with 3 at.% B at 673 K and annealed at 1673 K. (a) ADF STEM image acquired with collection angles of 48 - 200 mrad. (b) Overlapped B and C X-ray maps. (c) Si-K, (d) C-K and (e) Si-K X-ray maps. (f) Overlapped EDS spectra of the matrix (gray ) and precipitate (red). (g) ABF STEM image acquired with collection angles of 11 - 22 mrad. (h) B-K and Si-L2,3 EELS maps. (i) Overlapped EELS spectra of the matrix (blue) and precipitate (red). Convergence angle: 22 mrad.

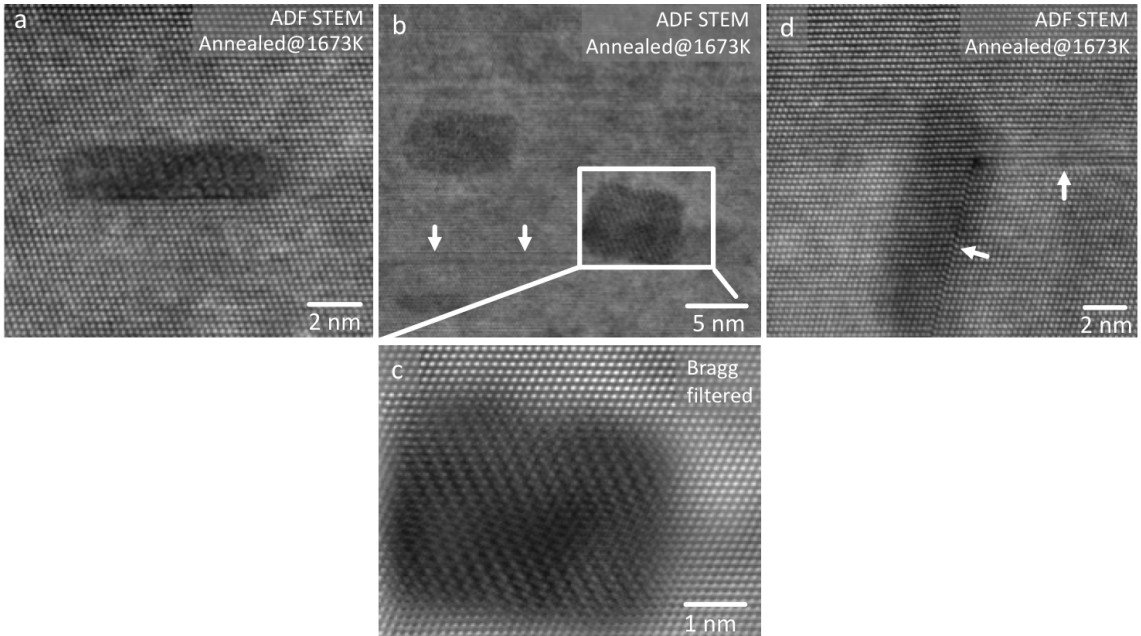

Figure 7: (a) to (d) ADF STEM image of a sample implanted with 3 at.% B at 673 K and annealed at 1673 K acquired with a convergence angle of 22 mrad and collection angles of 48 - 200 mrad. The arrows indicate planar defects in the SiC matrix. Zone axis: $[1\bar{1}0]_{\text{3C-SiC}}$.

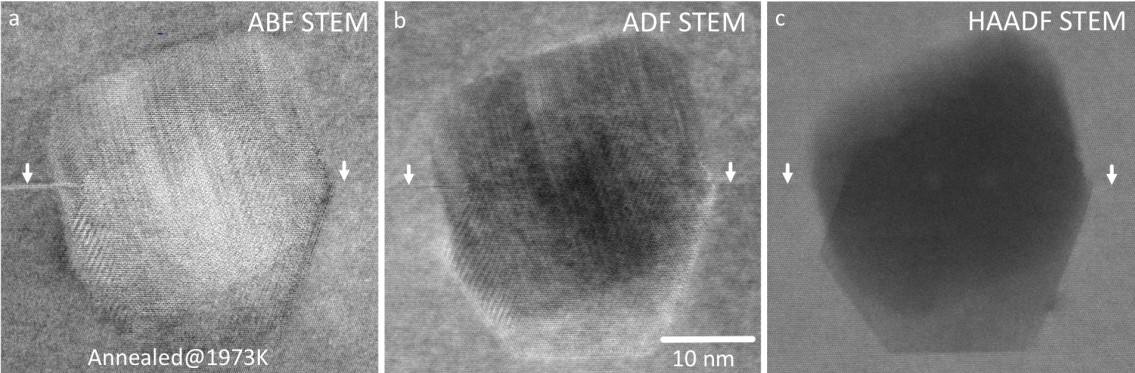

Figure 8: (a), (b) and (c) ABF, ADF and HAADF STEM images obtained simultaneously with a convergence angle of 22 mrad from a layer implanted with 2 at.% B at 673 K annealed at 1973 K. Collection angles for ABF: 11 - 22 mrad, ADF: 22 - 98 and HAADF: 99 - 200 mrad. The arrows point to a stacking fault in the matrix. Zone axis: $[1\bar{1}0]_{\text{3C-SiC}}$.

have been attributed to localized gap states and transitions between such states and the energy bands [34,35]. Specific characteristics of the boride determined from optical absorption, photoluminescence and charge transport data are: band gap of 2.09 eV, several disorder-induced intermediate gap states extending 1.2 eV above $E_v$, excitonic level at 1.56 eV above $E_v$, electron trap level around 0.27 eV below the bottom of the conduction band, and a conductivity of 20 $(\Omega \text{cm})^{-1}$ for the $B_{13}C_2$ stoichiometry at room temperature [34,36]. The typical random distribution of twins parallel to $\{1\bar{1}01\}_{B_{13}C_2}$ [30–32] is likely to contribute to the complex electronic configuration of this compound and to play a significant role in charge carrier recombination.

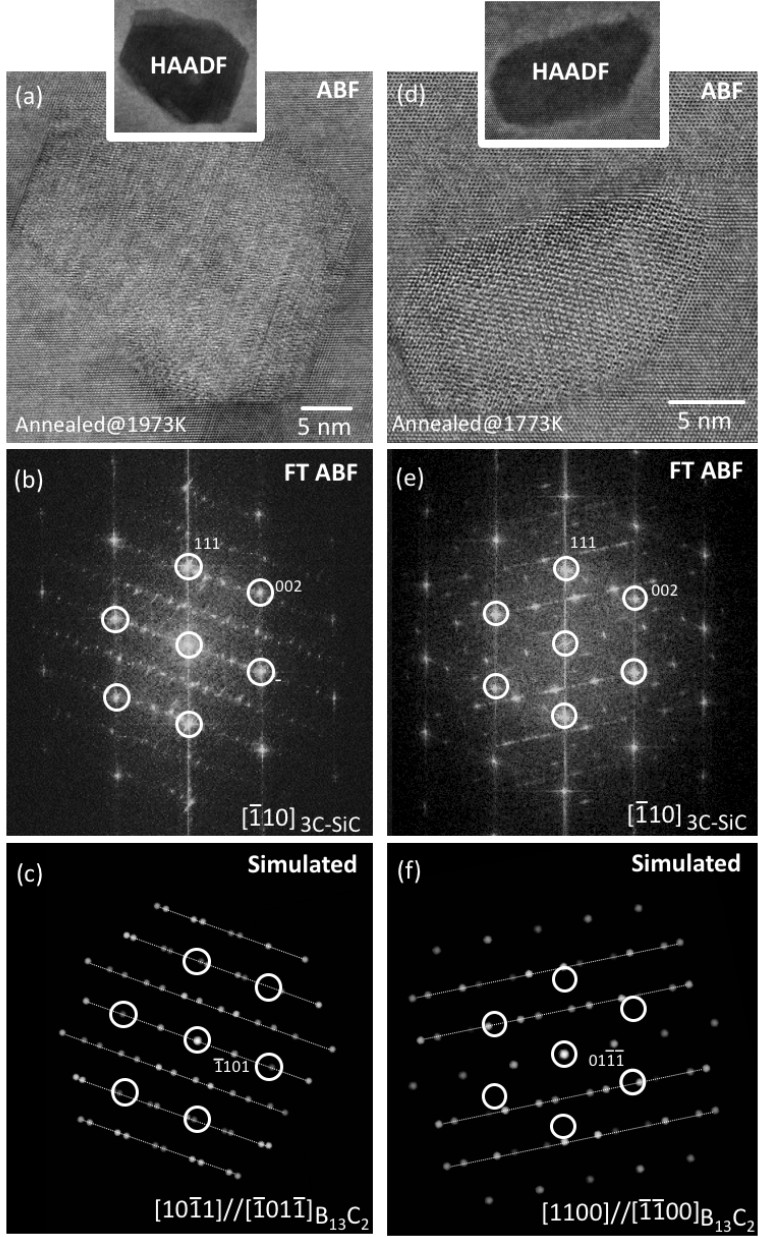

Figure 9: ABF and HAADF STEM images obtained simultaneously with a convergence angle of 22 mrad from a sample implanted with 2 at.% B at 673 K and annealed at (a) 1973 K and (d) 1773 K. Collection angles for ABF: 11 - 22 mrad, and HAADF: 99 - 200 mrad. (b) and (e) Fourier transforms of the images in (a) and (d), respectively. (c) and (f) Simulated diffraction patterns mirrored perpendicularly to $\left\{1\bar{1}01\right\}_{B_{13}C_2}$ and multiple scattering justify the additional spots and streaks observed in (b) and (e). Reflections of the 3C-SiC matrix are encircled for clarity.

## 3.3 Solubility of B in 3C-SiC

Changes in the concentration profiles compatible with long-range diffusion were only detected upon annealing at temperatures higher than 1773 K (see figure 10). At lower temperatures, the $B_{13}C_2$ precipitates trapped the B atoms and the low concentration of solute available in the 3C-SiC matrix prevented any significant long-range diffusion, i.e., low solubility led to stable concentration profiles, suggesting an apparent low diffusivity.

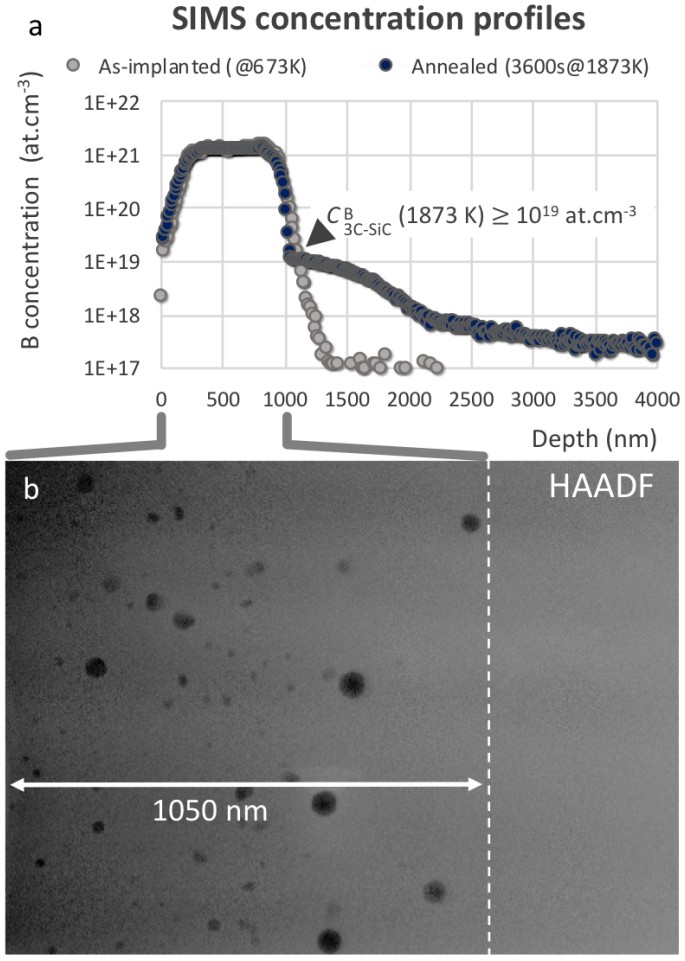

Figure 10: Concentration profile with 2 at.% B in as-implanted state and after annealing at 1873 K. (b) Cross section of the same sample as observed by HAADF with a convergence angle of 31 mrad and collection angles of 99 - 200 mrad.

Since the equilibrium solid solubility as a function of temperature, $C^{\mathrm{B}}_{\text{3C-SiC}}(T)$, is given by the concentration in thermodynamic equilibrium with the $B_{13}C_2$ phase, the concentration of free B immediately below the precipitation layer can be assumed $\leq C^{\mathrm{B}}_{\text{3C-SiC}}(T)$. Thus, the consistent absence of precipitates below $\sim 1000$ nm after annealing at 1873 K (see arrow in figure 10 (b)) enables to infer that $C^{\mathrm{B}}_{\text{3C-SiC}}(1873\mathrm{K}) \geq 10^{19}$ at.cm$^{-3}$ (see arrow in figure 10 (a)). This value is comparable to the solubility reported for 6H- and 4H-SiC at 1873 K [37,38], with $10^{20}$ at.cm$^{-3}$ proposed for these polytypes at the temperature of the 3C-SiC + $B_{13}C_2 \leftrightarrow$ L eutectic reaction $\sim 2500$ K [18–20]), which is expected to correspond to the equilibrium solid solubility limit.

In the present study, the low precipitate density and high dilutions achieved hindered an evaluation of solubility at temperatures higher than 1873 K. Precise determination of solubility at high temperatures is challenging for nanometric sources given the relatively poor lateral resolution of SIMS and the transient nature of the B concentration profiles [39], with fast diffusion due to steep gradients around the precipitates. In addition, precipitates stabilized by extended defects were present in layers implanted with 1 at.% B even after annealing at 2073 K for $1.4 \times 10^4$ s (see figure 5 (a)), although the B concentration measured at those depths ($\sim 200$ nm) was only $10^{17}$ at.cm$^{-3}$. Therefore, dissolution rather than diffusion may be the rate-controlling mechanism defining the concentration profiles at high temperatures, i.e., the 3C-SiC matrix may be rapidly depleted of B by fast diffusive fluxes in the neighborhood of

slowly dissolving $B_{13}C_2$ precipitates. In this case, the concentration measured immediately below the precipitation layer after relatively long heat treatments may, in fact, be significantly lower than the equilibrium solubility at the annealing temperature.

# 4   Conclusion

Annealing after implantation led to precipitation of $B_{13}C_2$ that trapped B up to 1773 K. Therefore, the low solubility induced stable concentration profiles and resulted in apparently low B diffusivity. The precipitates adopted the $(0001)_{B_{13}C_2}//\{111\}_{\text{3C-SiC}}$ and $\langle 1\bar{1}00\rangle_{B_{13}C_2}//\langle 1\bar{1}0\rangle_{\text{3C-SiC}}$ orientation relation with the matrix and exhibited planar defects on $\{1\bar{1}01\}_{B_{13}C_2}$. Strain generated by precipitate growth induced the formation of stacking faults and dislocations in 3C-SiC that contributed to lower the overall structural quality of the crystal. The presence of extended defects and precipitates may mask the presence of an IB in 3C-SiC and total elimination of these structures is challenging due to the locked configurations adopted. Correlation between SIMS concentration profiles and STEM observations enabled to infer that the solubility of B in perfect 3C-SiC crystals is $\geq 10^{19}$at.cm$^{-3}$ at 1873 K. Our results suggest that for IB formation, alternative methods for introducing high concentrations of B into 3C-SiC should be sought.

# Acknowledgements

The authors acknowledge the Norwegian Center for Transmission Electron Microscopy, NORTEM National Infrastructure (Grant agreement 197405/F50)

**Funding information**   The work reported here has been undertaken as part of the project SUNSiC - Efficient exploitation of the sun with intermediate band silicon carbide funded by the Research Council of Norway (Grant Agreement 229711/O20).

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
