# Peer review of "Boron-doping of cubic SiC for intermediate band solar cells: a scanning transmission electron microscopy study"

_SciPost Physics, doi:SciPost Phys. 5, 021 (2018)_

## Round 1 · Referee Report · Anonymous · 2018-5-22

**Referee's Report on "Boron-doping of cubic SiC for intermediate band solar cells: A scanning transmission electron microscopy study" by P.A. Carvalho, A. Thøgersen, Q. Ma, D.N. Write, S. Diplas, A. Galeckas, A. Azarov, V. Jokubavicius, M. Syväyärvi, B.G. Svensson, and O.M. Løvvik, submitted to SciPost Physics**

This manuscript presents a scanning transmission electron microscopy study of the annealing behavior of epitaxial 3C SiC heavily implanted with Boron (1 – 3 at..%). The concentration profile of the Boron was measured by SIMS. The principle finding is that implantation and annealing does not lead to isolated substitutional Boron at the high concentration levels required for application as an absorber in single-junction solar cells based on an intermediate band.

Since I have no previous experience with SciPost Physics, and in fact was not aware of its existence, I had a look at several recent issues as part of the process of deciding whether to accept the invitation to review this manuscript. It appears to me that most of the papers in this journal are on fashionable and/or esoteric subjects, and many are theoretical in nature. I did not notice any papers that would be regarded as materials science or characterization, so I'm surprised both that the authors chose to submit to this journal and that the editors are seriously considering it for publication. It does not appear to fit, and it's not clear whether others working in silicon carbide and/or solar cells will find it. However, I assume that the editors are willing to publish this manuscript, so I proceed to discuss it.

The work is sound and worthy of publication after the authors take into account the following comments and suggestions, listed in order of appearance.

Page 2: Experimental

The authors omit some information, which should be provided. They do not state the thickness, doping type and concentration of either the epilayer or substrate (Perhaps these are provided in [11]). They do not state the penetration depth of the Boron ions, although the TEM images and SIMS profiles suggest that all of the effects are within about a micron of the surface, so that an epilayer thickness of a few microns would be sufficient. How close is the ion fluence to that required to render the implanted layer amorphous? Because the implantations were performed at 773K, amorphization is not an issue. Figure 1 shows evidence for good periodicity before annealing.

Page 3

It may be helpful to provide details about the sample preparation for the electron microsopy.

Results and Discussion

Line 1: "STEM images obtained" → "Cross sectional STEM images obtained"

Line 5: "rather than from the introduction of the weakly-scattering B atoms" Presumably Carbon atoms are also displaced, but the same comment applies to C as to B.

Line 8: "B preferentially substitutes for Si" However, note that boron-nitrogen donor-acceptor pair spectra in 3C SiC indicate that there is a boron acceptor with an ionization energy of about 0.735 eV (A value of about 749 meV might be more reasonable). Since the DAP spectrum is "Type I" and nitrogen donors substitute for carbon, this boron also presumably substitutes for carbon. The story of boron in SiC is quite complicated. Relevant references for my comment are:

1) H. Kuwabara, S. Yamada and S. Tsunekawa, J. Lumin. 12-13, 531 (1976) and 2) J.W. Sun, I.G. Ivanov, S. Juillaguet and J. Camassel, Phys. Rev. B83, 195201 (2011).

Lines 8-10: "At high collection angles (98 – 200 mrad), no contrast between the implanted layers and the underlying material was discerned …" The lower part of Fig. 1(b) is a bit brighter, although there is no distinct boundary between bright and dark regions.

Page 4

Figure 1: Part (d) is fuzzy. The axes and tick marks (or grid lines) are barely discernable and must be made more bold.

Text: Line 1: "Magnified images of the region in Figure 1…" Would it be helpful to specify the position/depth? Does it matter, or is the implanted region uniform with respect to this measurement?

Line 14: What is meant by "apparent strain localization perpendicular to (111)"? Are you saying that there are localized regions of high strain which are needle-like with the axis perpendicular to a (111) face?

Page 6

Line 1: "a large density of stacking faults was observed on (111) planes parallel to the surface" For a reader who is not an expert in electron microscopy, it would be helpful to point out examples of such features on an image and/or describe their appearance.

Line 4: "(see Figure 2)" → "(see Figure 2 (a))"?

Figure 3 (c) is fuzzy, and the axes/grid lines are too light.

Note that there is lots of empty space surrounding most of the figures. The larger the figures, the more effective they will be. Efforts should be made, where possible, to enlarge the figures/images to use all of the available space.

Below Fig. 3, Line 8: A different B concentration was used for the sample annealed at 1873 K compared to the one annealed at 1773 K. Why? More generally, the reason for choosing three (at least) distinct boron concentrations should be stated.

Page 8, Figure 5: The images are very small, while there is much empty space available. Can something be done? Note that three distinct values of B concentration are represented in the images. Why? The caption might explain the purpose of the white arrows in the final image in part (a).

Page 9

Figure 9: In this case, some of the panels are so small that they are unreadable, although there is plenty of available space to make them bigger. Please use all of the available space to enlarge the images and panels, especially c, d, e, f.

Below Fig.6, Lines 4-5: Various types of brackets are used, associated with directions, lattice planes, etc. Are they all correct?

Page 11

Figure 9 Caption, Line 5: Should "(b) and (f)" be "(b) and (e)"?

Line 10: The "greater than or equal to" sign is rather subtle. I believe that it is associated with the largest boron concentration observed in the SIMS profile after annealing, highlighted by an arrow (which I failed to notice on first reading but do now). I suggest being a bit more explicit in the text.

Page 12: Figure 10

For part (a), make the axes/ticks/grid lines bolder.

In the caption for part (b), specify whether the cross-section image was obtained after annealing at 1873K.

References: Please check all of the references carefully, as there are quite a few minor errors.

[3]: "(IPV). Effect" → "(IPV) effect" would make more sense.

[5]: "*in SiC in Properties*" → "*in SiC* in *Properties*" (The second "in" should not be italicized.

[6]: I believe that the correct volume number is 93, not 3.

[9]: I believe that the correct journal is Phys. Rev. Lett., not Appl. Phys. Lett.

[17]: Should the '4' in "B4C" be a subscript?

[18]: Should the "x" in "BxC" be a subscript?

[18] and [20]: Unlike the other references, the year of publication is given before the page/article number. Please use a consistent set of conventions.

[29]: The "R" in "Radiation" should not be upper case, for consistency.

[31] and [33]: "App." → "Appl." is conventional.

---

## Round 3 · Author Response

Dear Editor and Reviewer,

The authors acknowledge the comments and suggestions and have carefully revised the manuscript taking them into account. Each comment/suggestion is addressed point by point below.

Reviewer’s comment 0: ‘…I did not notice any papers that would be regarded as materials science or characterization, I’m surprised both that the authors chose to submit to this journal and that the editors are seriously considering it for publication. ‘

Reply: SciPost has a section for ‘Solid State Physics – Experimental’ to which the paper has been submitted. The authors appreciate the double open access model of SciPost and wish to support and contribute to expand the experimental scope of SciPost.

Reviewer’s comment 1: ‘Experimental The authors omit some information, which should be provided. They do not state the thickness, doping type and concentration of either the epilayer or substrate (Perhaps these are provided in [11]). They do not state the penetration depth of the Boron ions, although the TEM images and SIMS profiles suggest that all of the effects are within about a micron of the surface, so that an epilayer thickness of a few microns would be sufficient.

Reply: The missing information has now been included in the 1st paragraph of the Experimental section in page 2 and a new author has been added. The penetration depth can be inferred from the B profiles and this information is given in lines 3-5 of the 1st paragraph of page 3.

Reviewer’s comment 2: How close is the ion fluence to that required to render the implanted layer amorphous? Because the implantations were performed at 773K, amorphization is not an issue. Figure 1 shows evidence for good periodicity before annealing.

Reply: Amorphization has been inferred from XRD measurements and Rutherford backscattered spectroscopy prior to TEM observation as is now stated in the last lines of the 1st paragraph of page 3.

Reviewer’s comment 3: Page 3 It may be helpful to provide details about the sample preparation for the electron microscopy.

Reply: Additional information has been included in the 2nd paragraph of page 3.

Reviewer’s comment 4: Results and Discussion Line 1: “STEM images obtained” -->“Cross sectional STEM images obtained”

Reply: The suggested change has been made in the 1st line of the Results and Discussion section (page 3).

Reviewer’s comment 5: Line 5: “rather than from the introduction of the weakly-scattering B atoms” Presumably Carbon atoms are also displaced, but the same comment applies to C as to B.

Reply: The text has been changed to reflect the idea (lines 3-4 of the 1st paragraph of page 4).

Reviewer’s comment 6: Line 8: “B preferentially substitutes for Si” However, note that boron-nitrogen donor-acceptor pair spectra in 3C SiC indicate that there is a boron acceptor with an ionization energy of about 0.735 eV (A value of about 749 meV might be more reasonable). Since the DAP spectrum is “Type I” and nitrogen donors substitute for carbon, this boron also presumably substitutes for carbon. The story of boron in SiC is quite complicated. Relevant references for my comment are: 2 1) H. Kuwabara, S. Yamada and S. Tsunekawa, J. Lumin. 12-13, 531 (1976) and 2) J.W. Sun, I.G. Ivanov, S. Juillaguet and J. Camassel, Phys. Rev. B83, 195201 (2011).

Reply: This important discussion has now been included in lines 5-12 of the 1st paragraph of page 4.

Reviewer’s comment 7: Lines 8-10: “At high collection angles (98 – 200 mrad), no contrast between the implanted layers and the underlying material was discerned …” The lower part of Fig. 1(b) is a bit brighter, although there is no distinct boundary between bright and dark regions.

Reply: The authors believe that the changes in HAADF images are associated with the gradual increase of thickness of the TEM sample. A comment has been inserted in the text to reflect this idea in lines 12 - 15 of the 1st paragraph of page 4).

Reviewer’s comment 8: Page 4 Figure 1: Part (d) is fuzzy. The axes and tick marks (or grid lines) are barely discernable and must be made more bold.

Reply: Figure 1 has been increased in size and the contrast of the graph in figure 1c has been enhanced. Hopefully it reads better now.

Reviewer’s comment 9: Text: Line 1: “Magnified images of the region in Figure 1…” Would it be helpful to specify the position/depth? Does it matter, or is the implanted region uniform with respect to this measurement?

Reply: The contrast variations were uniform across the implantation depth. A comment has been inserted to clarify this point (line 16 of the 1st paragraph of page 4).

Reviewer’s comment 10: Line 14: What is meant by “apparent strain localization perpendicular to (111)”? Are you saying that there are localized regions of high strain which are needle-like with the axis perpendicular to a (111) face?

Reply: The strain fields measured by GPA point to the compressive/tensile configurations typical of edge dislocations parallel to the viewing direction. A comment has been inserted in the 2nd paragraph of the Results and Discussion section in page 4) and yellow arrows are used to indicate the features in figure 2c.

Reviewer’s comment 11: Page 6 Line 1: “a large density of stacking faults was observed on (111) planes parallel to the surface” For a reader who is not an expert in electron microscopy, it would be helpful to point out examples of such features on an image and/or describe their appearance.
Reply: Additional clarification has been included in the 3rd paragraph of the Results and Discussion section in page 4 and in the caption of figure 3.

Reviewer’s comment 12: Line 4: “(see Figure 2)” -->“(see Figure 2 (a))”? Figure 3 (c) is fuzzy, and the axes/grid lines are too light. Note that there is lots of empty space surrounding most of the figures. The larger the figures, the more effective they will be. Efforts should be made, where possible, to enlarge the figures/images to use all of the available space.

Reply: The figures have been enlarged to use the available space.

Reviewer’s comment 13: Below Fig. 3, Line 8: A different B concentration was used for the sample annealed at 1873 K compared to the one annealed at 1773 K. Why? More generally, the reason for choosing three (at least) distinct boron concentrations should be stated.

Reply: Combinations of different B concentration level and temperature and time of annealing have been investigated to infer precipitation and defect formation trends within a large window of processing parameters. This clarification has been added in the Experimental section (lines 6-8 of the 1st paragraph of page 3) as well as in the Results and Discussion section (1st paragraph of page 7)

Reviewer’s comment 14: Page 8, Figure 5: The images are very small, while there is much empty space available. Can something be done? Note that three distinct values of B concentration are represented in the images. Why? The caption might explain the purpose of the white arrows in the final image in part (a).

Reply: Figure 5 has been enlarged. Please refer to the Reply to comment 13 regarding the different B compositions. The purpose of the arrows is now explained in the caption.

Reviewer’s comment 15: Page 9 Figure 9: In this case, some of the panels are so small that they are unreadable, although there is plenty of available space to make them bigger. Please use all of the available space to enlarge the images and panels, especially c, d, e, f.

Reply: Figure 9 has been enlarged.

Reviewer’s comment 16: Below Fig.6, Lines 4-5: Various types of brackets are used, associated with directions, lattice planes, etc. Are they all correct?

Reply: The brackets have been verified and one instance was corrected

Reviewer’s comment 17: Page 11 Figure 9 Caption, Line 5: Should “(b) and (f)” be “(b) and (e)”?

Reply: It should have been (e) and the mistake is now corrected.

Reviewer’s comment 18: Line 10: The “greater than or equal to” sign is rather subtle. I believe that it is associated with the largest boron concentration observed in the SIMS profile after annealing, highlighted by an arrow (which I failed to notice on first reading but do now). I suggest being a bit more explicit in the text. Page 12: Figure 10 For part (a), make the axes/ticks/grid lines bolder. In the caption for part (b), specify whether the cross-section image was obtained after annealing at 1873K

Reply: The arrow is now explicitly mentioned in the text (last paragraph of page 9) and in the caption of Figure 10. The image has been enlarged and hopefully reads better now. The cross-section image was obtained in the same sample after annealing at 1873 K and this is now explicitly stated in the caption.

Reviewer’s comment 19: References: Please check all of the references carefully, as there are quite a few minor errors. [3]: “(IPV). Effect” -->“(IPV) effect” would make more sense. [5]: “in SiC in Properties” -->“in SiC in Properties” (The second “in” should not be italicized. [6]: I believe that the correct volume number is 93, not 3. [9]: I believe that the correct journal is Phys. Rev. Lett., not Appl. Phys. Lett. [17]: Should the ‘4” in “B4C” be a subscript? [18]: Should the “x” in “BxC” be a subscript? [18] and [20]: Unlike the other references, the year of publication is given before the page/article number. Please use a consistent set of conventions. [29]: The “R” in “Radiation” should not be upper case, for consistency. [31] and [33]: “App.” -->“Appl.” is conventional.

Reply: The references have been thoroughly checked and corrected where necessary.

Sincerely,
Patricia Almeida Carvalho

---

## Round 3 · List of Changes

The changes relative to the reviewer comments/suggestions are presented in red text in the revised version (v2).

The figures have been increased in size and the contrast of figure 1 (c) is enhanced.

Yellow arrows indicate apparent compressive/tensile stress fields in figure 2 (c)

---

## Editorial Decision

published